# Functional SNPs in the Human Autoimmunity-Associated Locus 17q12-21

**DOI:** 10.3390/genes10020077

**Published:** 2019-01-23

**Authors:** Alina S. Ustiugova, Kirill V. Korneev, Dmitry V. Kuprash, Marina A. Afanasyeva

**Affiliations:** 1Engelhardt Institute of Molecular Biology, Russian Academy of Sciences, 119991 Moscow, Russia; ustugovaalina@yandex.ru (A.S.U.); kirkorneev@yandex.ru (K.V.K.); kuprash@gmail.com (D.V.K.); 2Biological Faculty, Lomonosov Moscow State University, 119234 Moscow, Russia

**Keywords:** enhancer, regulatory variant, luciferase reporter assay, post-GWAS, multiple sclerosis, ulcerative colitis, Crohn’s disease, type 1 diabetes, rheumatoid arthritis, primary biliary cirrhosis

## Abstract

Genome-wide association studies (GWASes) revealed several single-nucleotide polymorphisms (SNPs) in the human 17q12-21 locus associated with autoimmune diseases. However, follow-up studies are still needed to identify causative SNPs directly mediating autoimmune risk in the locus. We have chosen six SNPs in high linkage disequilibrium with the GWAS hits that showed the strongest evidence of causality according to association pattern and epigenetic data and assessed their functionality in a local genomic context using luciferase reporter system. We found that rs12946510, rs4795397, rs12709365, and rs8067378 influenced the reporter expression level in leukocytic cell lines. The strongest effect visible in three distinct cell types was observed for rs12946510 that is predicted to alter MEF2A/C and FOXO1 binding sites.

## 1. Introduction

The study of human 17q12-21 locus began with the discovery of its association with inheritable predisposition to breast cancer [1]. Later, strong association with asthma was also discovered [2]. Since the identification of the first single-nucleotide polymorphisms (SNPs) in the locus that strongly correlate with susceptibility to childhood asthma and *ORMDL3* gene expression level [3], 17q12-21 has been the most replicated and most significant asthma locus defined by genome-wide association studies (GWASes) and meta-analyses of GWASes, and the list of candidate genes was substantially extended [4]. More recently, several GWASes also mapped autoimmunity association signals to 17q12-21 for primary biliary cirrhosis [5,6], rheumatoid arthritis [7], type I diabetes [8], ulcerative colitis [9], and Crohn’s disease [10]. 17q12-21 has also been linked to allergy [11]. Interestingly, some variants have opposing risk alleles for autoimmunity and asthma. Taken together, these data highlight 17q12-21 as an important locus for immune system in general.

Autoimmunity-associated SNPs in the 17q12-21 locus overlap with several genes: *NEUROD2*, *PPP1R1B*, *STARD3*, *TCAP*, *PNMT*, *PGAP3*, *ERBB2*, *MIR4728*, *MIEN1*, *GRB7*, *IKZF3*, *ZPBP2*, and *GSDMB.* The role of these genes in the immune system is not obvious, with the exception of *IKZF3* that encodes Aiolos transcription factor and is particularly important for normal development and function of B cells [12]. Considering also the fact that regulatory SNPs may influence gene expression over very long distances, identification of causative variants and their target genes becomes an important task on our way to deeper understanding of molecular mechanisms of autoimmunity. While the target genes can be discovered using genome editing, the true causative SNPs in the locus must be identified first.

Autoimmune diseases arise due to intrinsic imperfection and defects in the mechanisms of immunological tolerance. Development of these complex diseases depends on multiple genetic and environmental factors. Pathological process likely begins with activation of innate immunity after recognition of self or non-self-molecules such as nucleic acids, followed by inflammation and activation of self-reactive T and B cells which are normally present in every individual [13]. Various cell types can participate in the pathogenesis of an autoimmune disease, including numerous immune cell subsets as well as non-immune cells of an affected organ. However, comprehensive knowledge of the immune cell types mainly involved in the development of a particular disease, as well as those, mediating inheritable disease risk, is still missing. These may include B cells, Th17, Th1, Treg, monocytes, dendritic cells, neutrophils, and many others. The picture becomes even more complex if numerous cell subsets and states are considered [14]. That is why functional follow-up studies of potentially causative SNPs should include as much relevant cell types as possible.

Identification of causative variants includes two stages: fine-mapping of the associated locus and experimental testing of the resulting candidate SNPs. A variety of works have been published to date that use statistical methods, functional genomic annotations, and expression data to determine which variants are most likely to be functional [15]. However, laboratory follow-up studies are scarce. Here we used fine-mapping results of Farh et al. [16] and Schmiedel et al. [17] to select the most probable causative autoimmunity-associated SNPs in the human 17q12-21 locus for experimental validation. We studied their effect on transcription in the luciferase reporter system which is most suited for analysis of one or few loci and enables detection of regulatory activities of long sequences that depend on cooperative binding of different transcription factors [18]. We found four SNPs that significantly influenced the reporter expression level in lymphoid and monocytic cell lines.

## 2. Materials and Methods

### 2.1. SNPs Selection

We used the results of fine-mapping of autoimmunity-associated loci performed by Farh et al. [16] and of asthma-associated 17q12-21 SNPs performed by Schmiedel et al. [17]. The list of candidate causal SNPs for 39 immune and non-immune diseases and enhancer annotations from the former work was downloaded from the data portal [19]. For each autoimmune GWAS hit in the dense 17q12-21 autoimmune SNP cluster (GRCh37/hg19 chr17:37740161-38066240) we picked a candidate with the highest PICS (probabilistic identification of causal SNPs) probability (PP) among those that overlap any immune enhancer. Of 136 polymorphisms analyzed by Schmiedel et al., we chose SNPs highlighted by the authors for their overlap with DNase I hypersensitivity sites highly specific for lymphocytes (Figure 3a in [17]).

### 2.2. Publication Mining for Expression Quantitative Trait Loci (eQTL)

The results of 16 studies [20,21,22,23,24,25,26,27,28,29,30,31,32,33,34,35] on expression quantitative trait loci (eQTL) in human peripheral blood, primary leukocytes, and lymphoblastoid cell lines were analyzed. We extracted all eQTL with false discovery rate < 0.05 for genes located within 100kb distance from our candidate variants and obtained r^2^ values for linkage between our candidates and these eQTL using SNAP Pairwise LD tool [36]. If strong linkage was found (r^2^ > 0.8), we reported the strongest association with the corresponding eQTL among our candidate variants.

### 2.3. Enhancers Cloning

Five putative enhancer sequences were amplified by PCR using genomic DNA isolated from a healthy donor peripheral blood using Genomic DNA Purification Kit (Thermo Fisher Scientific, Waltham, MA, USA) with the following primers introducing restriction sites (lowercase letters) for BamHI (forward, except BglII for rs12709365/rs13380815) and SalI (reverse) (Table 1).

To create size-matched negative controls for experimental vectors, we amplified genomic segments of similar sizes not carrying any epigenetic traits of active enhancers or repressors in leukocytes using primer sequences shown in Table 2.

Putative enhancers and control sequences were inserted into pGL3-basic vector (Promega, Fitchburg, WI, USA) downstream of the luciferase gene by the BamHI/SalI restriction sites. Synthetic promoter from the pGL4.24 vector (Promega, Fitchburg, WI, USA) was inserted upstream of the luciferase gene by the HindIII/NcoI restriction sites.

### 2.4. PCR Mutagenesis

After sequencing of cloned enhancers alternative SNP alleles were introduced by overlap extension PCR using the following primers (Table 3).

Half-fragments were first separately amplified with corresponding overlap and terminal primer pairs from the plasmid template. After gel-purification, 50 ng of each half-fragment were used for the final 20 μL joining reaction with the Phusion high-fidelity DNA polymerase (Thermo Fisher Scientific, Waltham, MA, USA) by the following protocol: initial denaturation 98 °C—1 min; 5 cycles: 98 °C—20 s, Tm of the overlapping segment—20 s, 72 °C—30 s/kb length of the longer half-fragment; pause at 72 °C and adding the terminal primers (1 μM); 35 cycles: 98 °C—10 s, Tm of the terminal primers—20 s, 72 °C—30 s/kb of the full enhancer; final extention at 72 °C—7 min. The resulting mutated enhancer sequences were inserted into the reporter vector using BamHI/SalI restriction sites. All constructs were verified by Sanger sequencing.

### 2.5. Cells

Jurkat and MT-2 were obtained through the NIH AIDS Research and Reference Reagent Program. U-937 was kindly provided by Dr. V.S. Prassolov (Engelhardt Institute of Molecular Biology, Moscow, Russia). MP1 and Nalm6 were kindly provided by Dr. Edward A. Clark (University of Washington, USA). All cells were cultured in RPMI 1640 medium (Paneco, Moscow, Russia) supplemented with 10% FBS (Biosera, Nuaille, France), 2 mM L-glutamine (Gibco™, Thermo Fisher Scientific, Waltham, MA, USA), 100 U/mL Penicillin, 100 μg/mL Streptomycin (Gibco™, Thermo Fisher Scientific, Waltham, MA, USA) in humidified atmosphere at 37 °C, 5% CO_2_.

### 2.6. Activation of U-937

The protocol of activation was developed previously [37]. Cells were seeded in a culture-treated flask at 0.8 mln/mL density and 10 ng/mL of phorbol 12-myristate 13-acetate (Sigma-Aldrich, St. Louis, MO, USA) was added to culture medium. Twenty-four hours later 1 μg/mL of lipopolysaccharide (LPS; E. coli O111:B4, L2630, Sigma-Aldrich, St. Louis, MO, USA) was also added for another 3 h. After electroporation, cells were transferred to fresh medium containing 1 μg/mL LPS.

### 2.7. Luciferase Reporter Assay

Cells were transfected using Neon™ Transfection System (Thermo Fisher Scientific, Waltham, MA, USA) with the following parameters (Table 4).

For two million cells in 100 μL of buffer R we added 0.5 μg of pRL-CMV *Renilla* luciferase control vector (Promega, Fitchburg, WI, USA), 1.2 pmol of the pGL3-based test vector, and salmon sperm DNA (Sigma 31149) to the total DNA amount of 26.5 μg. Luciferase activity was measured 24 h after electroporation using Dual-Luciferase Reporter Assay System (Promega, Fitchburg, WI, USA), Hidex Bioscan Plate Chameleon Luminometer and MicroWin 2000 software. Minimum of two independent experiments were performed, each including three to four technical replicates.

### 2.8. Statistical Analysis

Statistical significance of the difference in the luciferase signal between alternative SNP alleles was determined using two-tailed unpaired Student’s *t*-test. Calculation was performed in Microsoft Excel. Data were represented as mean ± SEM.

## 3. Results

### 3.1. Selection of 17q12-21 SNPs for Experimental Validation

A mathematical algorithm developed by Farh et al., called PICS [16], assigns each variant a probability of being causative. We chose the following SNPs based on their analysis of autoimmunity-associated polymorphisms: rs12946510 for multiple sclerosis (PP = 0.314); rs12709365/rs13380815 (these two variants are tightly linked with r^2^ = 1) for ulcerative colitis (PP = 0.0408), Crohn’s disease (PP = 0.0382), and rheumatoid arthritis (PP = 0.0358); and rs8067378 for primary biliary cirrhosis (PP = 0.1079) (for details of selection process see Materials and Methods section).

Schmiedel and co-authors [17] used epigenetic data to analyze 136 asthma-associated SNPs in the 17q21 locus and found that three of them overlapped DNase I hypersensitivity sites highly specific for lymphocytes: rs12946510 (already included in our list), rs2313430, and rs4795397. The latter two appeared to be also associated with ulcerative colitis as was shown in a GWAS not covered by Farh et al. [38]. All candidate SNPs are in strong linkage disequilibrium (LD) with each other in European ancestry (Figure 1b). Short summary of the selected candidate SNPs is present in Table 5. More details can be found in Appendix A.

### 3.2. Possible Target Genes of the Selected Candidate SNPs

In our previous works we studied influence of regulatory SNPs on the promoter activity of the nearest genes as they were the most probable targets for the studied variants. However, functional polymorphisms do not always regulate expression of the nearest gene [25,45,46,47,48,49]. A number of efforts were made to identify eQTL in several human cell types and tissues, including whole blood, lymphoblastoid cell lines, and some major populations of primary leukocytes [20,21,22,23,24,25,26,27,28,29,30,31,32,33,34,35,50]. These data have shown that regulatory SNPs can indeed be located several million bases away from their target genes or even at another chromosome, and that transcriptional regulation they exert is often specific for cell type and functional state of the cells (for example, activation mode). We performed a publication search for eQTL in primary leukocytes, whole blood, and lymphoblastoid cell lines, and found evidence that selected SNPs may be associated with the expression levels of at least eight genes within the 100 kb distance (Table 6). Therefore, we decided to use a minimal synthetic promoter to study the influence of selected SNPs on enhancer activity in the luciferase reporter assay.

### 3.3. Effects of the Selected Variants on Transcription in Luciferase Reporter Assay

We cloned putative enhancers containing selected SNPs into luciferase reporter vector to test how each variant could affect enhancer properties of the surrounding genomic DNA in a range of human leukocytic cell lines. As gene targets of the selected SNPs are not precisely known, we used an artificial minimal promoter from the pGL4.24 vector. After determining the likely borders of enhancers surrounding each SNP (Figure 1a), corresponding DNA fragments were amplified from human genomic DNA and inserted downstream of the luciferase gene. Alternative alleles were introduced by overlap extension PCR.

To maximize the chance to observe the putative enhancers work, we used a variety of human leukocytic cell lines for transfection (Table 7).

The results of the luciferase assay are shown in Figure 2. As alternating alleles have similar frequencies in European population for all the studied SNPs, we refer to them as risk and non-risk instead of major and minor alleles.

For rs12946510, the difference in luciferase expression between two allelic variants was statistically significant in both B cell lines and activated U-937. Direction of the effect was cell-dependent. Risk allele inhibited enhancer activity in mature B cell line MP1, contrary to pre-B cell line Nalm6 and activated U-937 where it boosted luciferase expression.

Among two candidate SNPs linked to the index variant rs2305480, only rs4795397 showed weak but statistically significant influence on luciferase activity. Its risk allele corresponded to lower luciferase signal in MP1 cells.

In another pair of tightly linked candidates, rs12709365 and rs13380815, associated with the same GWAS hit, only rs12709365 showed statistically reliable inhibition of luciferase expression by the risk allele in activated U-937.

Finally, risk allele of rs8067378 corresponded to twice higher reporter signal in MT-2 cell line.

We next analyzed in silico which transcription factors could differentially bind to alternative alleles of the studied SNPs using PERFECTOS-APE software [63,64] with JASPAR collection of binding motifs [65] (Appendix A). The most potent polymorphism among our candidates—rs12946510—was predicted to alter binding affinity of five transcription factors to the corresponding DNA more than 20-fold: MEF2A, SOX3, MEF2C, FOXO1, and SOX6. Of them, MEF2C and FOXO1 are highly expressed in MP1 cell line, while SOX6 and MEF2A are scarce and SOX3 is totally absent [66]. ChIP-seq data from ENCODE is available for MEF2A and MEF2C, and their signals overlap rs12946510 in the lymphoblastoid cell line GM12878 (Figure 3). Therefore, in MP1 cells rs12946510 could act through creating MEF2A/MEF2C binding site or by disrupting the binding site for FOXO1.

## 4. Discussion

When autoimmunity-associated SNPs from the GWAS Catalog [67] are visualized in the Genome Browser [51,52], a dense cluster can be seen in the human 17q12-21 locus. This area contains hundreds of SNPs in strong LD (r^2^ > 0.8), and each can potentially be causative, i.e., underlie the molecular mechanism that modulates disease risk. Several approaches exist to narrow down the list of candidate causative variants (fine-mapping) having their own strengths and weaknesses, including simple heuristic methods, penalized regression, Bayesian methods, trans-ethnic fine-mapping, genomic annotation, and integration of gene expression data with GWAS data [15,68].

We used the results of two fine-mapping studies. Farh and colleagues [16] developed PICS algorithm that makes use of densely-mapped genotyping data to estimate each SNP’s probability of being a causal variant given the observed pattern of association at the locus and combined this excellent mathematical tool with experimental data on histone modifications in primary human cells. Schmiedel et al., analyzed 136 asthma-associated SNPs in the 17q12-21 locus for their coincidence with DNase hypersensitivity sites in 62 primary cell types [17]. Of the six predicted autoimmunity-associated functional SNPs in the 17q12-21 locus, four showed ability to alter reporter expression level in our experimental system: rs12946510, rs4795397, rs12709365, and rs8067378—one for each index SNP.

Polymorphism rs12946510 showed the brightest results in our experimental follow-up. The highest difference between alternative alleles was observed for this SNP and it was the only candidate which revealed its functionality in several cell types. This variant was described in detail by Farh et al., as an example of a polymorphism with bright genetic and epigenetic evidence of causality which simultaneously is an eQTL [16]. Its risk allele is associated with decreased *IKZF3* expression in peripheral blood [24]. *IKZF3* encodes Aiolos transcription factor which is first detected at low levels in pro-B cells and is upregulated in pre-B and mature peripheral B cells. Aiolos is necessary for normal B cell development and generation of long-lived, high-affinity bone marrow plasma cells [12]. In our reporter system, risk allele of rs12946510 also inhibited transcription in mature B cell line MP1. In subjects carrying the risk allele of rs12946510 this variant is likely to decrease enhancer activity in mature B cells resulting in lower Aiolos expression and impaired B cell function. This, in turn, can lead to autoimmunity through spontaneous production of autoantibodies or other mechanisms [69].

An opposing effect of rs12946510 on luciferase expression was observed in Nalm6 and activated U-937, whose primary counterparts (pre-B cells and activated monocytes respectively) are not normally present in peripheral blood. Given that promoter-enhancer pairs are often cell-type specific [70], the question which genes in these cells may be sensitive to rs12946510 requires further investigation.

Most likely, in MP1 cell line rs12946510 acts through creating MEF2C binding site or by disrupting the binding site for FOXO1. MEF2C is B cell specific in mice [71]. It orchestrates lineage commitment and is needed for normal development and functioning of B cells [72]. FOXO1 is another transcription factor important for B cell development. It promotes and stabilizes specification to the B cell lineage [73].

Two possible explanations can be proposed for divergent effect of rs12946510 on enhancer activity in MP1 (inhibition by the risk allele) and Nalm6/U-937 (activation by the risk allele). This divergence is possible if the allelic effect is mediated through altered MEF2C binding in both cases, as MEF2C was previously shown to be either activating or repressing depending on its protein partners [72]. Alternatively, bidirectional effect observed in different cell types could be explained by effective predominance of either one or another transcription factor which binding sites are altered by rs12946510, because risk allele creates MEF2C binding motif while disrupting it for FOXO1.

Two candidate polymorphisms associated with the same GWAS hit, rs13380815 and rs12709365, could not be differentiated by fine-mapping strategy based on genomic annotation, because they are separated by only 183 nucleotides on the chromosome and overlap the same epigenetic marks. They cannot be prioritized by mathematical, trans-ethnic, or gene expression approaches ether, due to absolute LD (r^2^ = 1) in all populations studied by the 1000 Genomes Project. Only experimental testing enabled to pinpoint one of them—rs12709365—as a possible causative SNP for inflammatory bowel disease and rheumatoid arthritis.

Fine-mapping strategy used by Schmiedel et al. [17] could not discriminate between two possibly functional SNPs, rs2313430 and rs4795397, associated with the same index SNP. In contrast, PICS algorithm, which takes into account the statistical data and linkage structure of the locus, assigns to these two variants very different probabilities of being causative: 0.0006 for rs2313430, and 0.0113 for rs4795397. In line with this prediction, rs4795397 showed statistically significant effect on luciferase expression. However, this data should be taken with caution as the surrounding DNA sequence did not demonstrate any enhancer properties in our system despite strong H3K27Ac and DHS signals in primary immune cells.

Polymorphism rs8067378 is an index SNP for primary biliary cirrhosis and at the same time it has the highest probability of being causative among its LD group members according to PICS. Verlaan et al., mentioned it as one of the top three allelic expression-associated SNPs in the locus, observed in both the CEU (Utah Residents with Northern and Western European Ancestry) and YRI (Yoruba in Ibadan, Nigeria) populations. These authors also found allelic differences in protein binding for this polymorphism using electrophoretic mobility shift assay, however they could not detect any allelic effect in luciferase reporter system [74]. This discrepancy between our results probably comes from a different study design used by Verlaan et al., because a three-times shorter genomic segment was used as an enhancer which did not improve expression from the SV40 promoter, and only one leukocytic cell line was used for transfection (Jurkat) in which we did not detect any significant effect either.

Identification of causative polymorphisms in each GWAS locus is the first step towards our understanding of how non-coding genetic variation translates into disease. This understanding can be built through identification of intermediate phenotypes, such as gene expression and functional differences between immune cells bearing alternative alleles. Studying narrow sub-populations of primary cells from donors of different genotypes can provide useful insights, however addressing the question of causal relationships between intermediate phenotypes and disease requires additional approaches, including Mendelian randomization and longitudinal studies [14]. Genetic engineering can also help to address this point without a need for huge sample sizes if adequate ex vivo cell systems are used.

In conclusion, using systematic approach we found four autoimmunity-associated SNPs in the human 17q12-21 locus with strong statistical, epigenetic and functional evidence of causality. Their target genes in various cell types are of great interest and can be further identified using genome editing coupled with transcriptomics.

## Figures and Tables

**Figure 1 genes-10-00077-f001:**
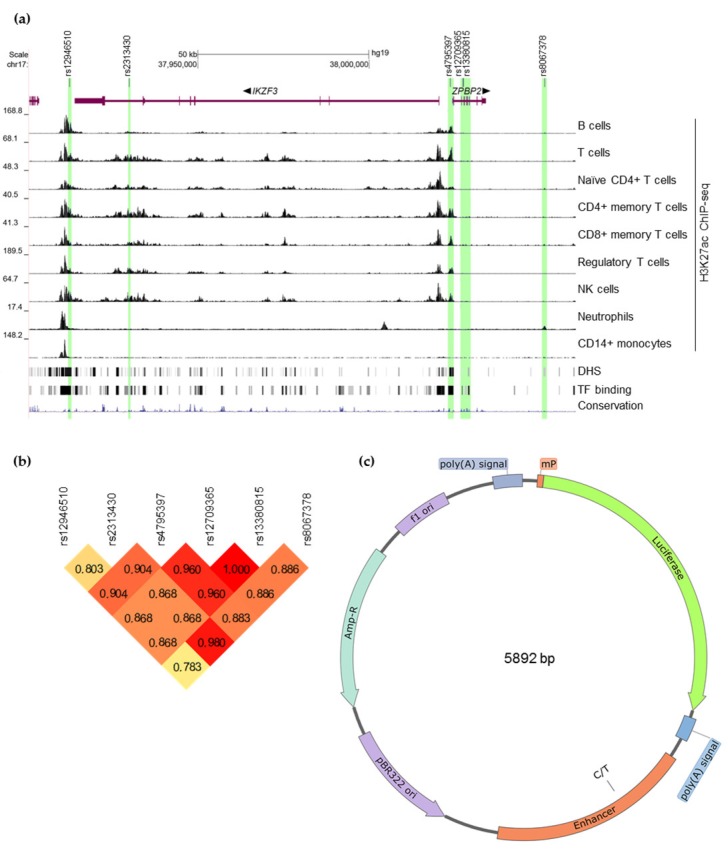
Candidate functional SNPs in the human autoimmunity-associated locus 17q12-21 and the corresponding putative enhancers. (**a**) University of California Santa Cruz (UCSC) Genome Browser [51,52] view of the candidate functional SNPs (black lines) associated with autoimmune diseases in the 17q12-21 locus and the corresponding putative enhancers (highlighted in green). Histone acetylation tracks provided by ENCODE [53,54] illustrate enhancer activity in a variety of primary human leukocytes. Gene schemes are shown in purple. Thin lines correspond to introns, thicker parts correspond to exons with coding sequences being the thickest. Direction of transcription is indicated by arrows next to gene names. DHS—DNaseI hypersensitivity clusters in 125 cell types from ENCODE (V3). TF binding—ChIP-seq peaks for 161 transcription factors in 91 cell types from ENCODE. (**b**) Linkage disequilibrium between candidate SNPs (r^2^ in CEU (Utah Residents with Northern and Western European Ancestry) population according to 1000 Genomes Phase 3 [44,55]. (**c**) The design of reporter vectors visualized using SnapGene software (from GSL Biotech; available at snapgene.com). The construct used for rs12946510 is shown as an example. Genomic segments highlighted in (a) were inserted downstream of the luciferase gene placed under the synthetic minimal promoter (*mP*) from the pGL4.24 vector. Two allelic variants were created to test each candidate SNP.

**Figure 2 genes-10-00077-f002:**
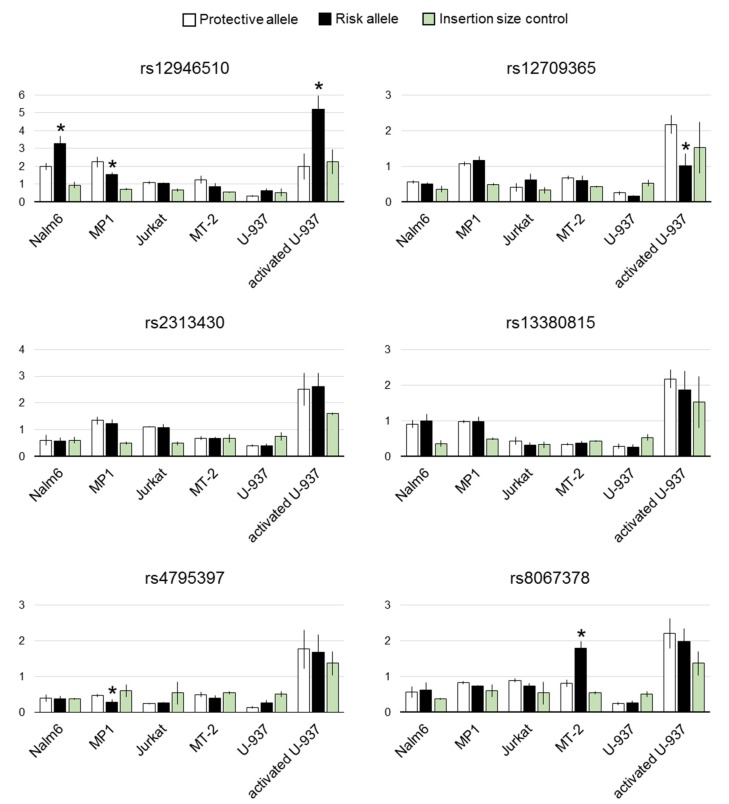
Effects of the candidate functional SNPs on enhancer activity in luciferase reporter assay. Two allelic variants of putative enhancers containing each tested SNP were cloned into modified pGL3 vector downstream of the firefly luciferase gene. A panel of leukocytic cell lines were transfected with these reporter constructs together with pRL. Luminescence was measured 24 h after. Relative firefly to *Renilla* signal was normalized to the value obtained from the promoter-only control. White boxes—protective alleles; black boxes—risk alleles; green boxes—size-matched negative controls. N ≥ 6. Mean ± SEM. * p < 0.05 for the difference between alternative alleles.

**Figure 3 genes-10-00077-f003:**
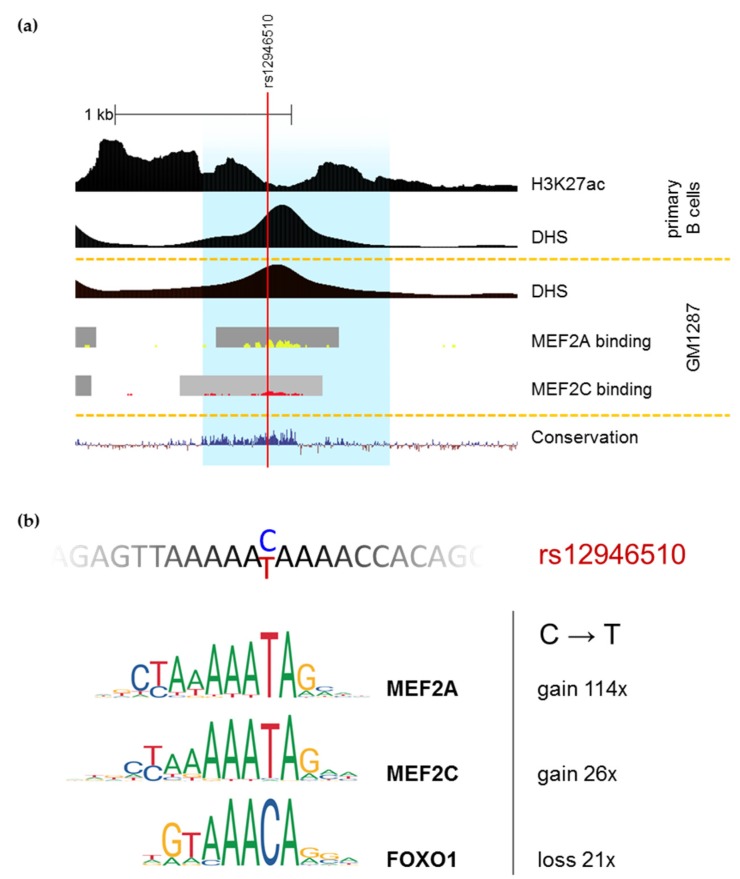
Polymorphism rs12946510 alters MEF2A/C and FOXO1 binding sites. (**a**) University of California Santa Cruz (UCSC) Genome Browser [51,52] view of the putative enhancer surrounding rs12946510 (highlighted in blue) with selected ENCODE [53,54] regulation tracks and conservation among 100 vertebrates. Histone modification H3K27ac is indicative of an active enhancer and shows a local drop in the area of transcription factors binding. DNase I hypersensitivity (DHS) density signal corresponds to open chromatin and shows similar profile in primary cells and lymphoblastoid cell line GM12878, for which ChIP-seq signals of MEF2A and MEF2C binding are available. (**b**) DNA sequence surrounding rs12946510 and ChIP-seq based motif logos for MEF2A, MEF2C, and FOXO1 according to JASPAR 2018 [65]. C to T substitution dramatically alters motif P-value for MEF2A and moderately for MEF2C and FOXO1 (fold-change is given according to PERFECTOS-APE [63,64]).

**Table 1 genes-10-00077-t001:** Primer sequences used for enhancer cloning.

SNP	Enhancer (Chr17 Hg19 Coordinates)	Length, bp	Forward Primer (5′-3′)	Reverse Primer (5′-3′)
rs12946510	37912040- 37913027	1057	TTTggatccAATGCAATTCCAGTGGGGGT	TTTgtcgacGCCCCTCAGTAGCTGGTTTT
rs2313430	37929651- 37930102	718	TTTggatccGGACATCAGGCCTTTGGGAA	TTAgtcgacTGAACTGGGGAAGAGGGACA
rs4795397	38023183- 38024680	1705	AAAggatccTGGTTAAGTCTCCTCTCATAGGATT	AAAgtcgacTAAAACGCGGGCATTGGACT
rs12709365/rs13380815	38026822- 38029614	3029	TTTagatctAAGGACTTCAGACGAGCGTT	TTTgtcgacTACTCCAGCTCTCTTTTGAGAA
rs8067378	38050770- 38051866	1448	TTTggatccCACCTGCTCCTGTCTGATGC	TTTgtcgacGAGCCCATTGCAAGCAGTCT

**Table 2 genes-10-00077-t002:** Primers used for the cloning of the size-matched control sequences.

Control For	Hg19 Coordinates	Length, bp	Forward Primer (5′-3′)	Reverse Primer (5′-3′)
rs2313430	chr17:38087769- 38088481	713	TTTggatccATACAGTGATTGCATTTGCTTCG	TTTgtcgacTGATCATCGCCATCTTCATTTACTT
rs12946510	chr10:6101364- 6102272 *	909	AAggatccGCTGTACCCAGTGCGTAG	TATgtcgacTACTGCAAAGTGGCTATGAAG
rs4795397/rs806737	chr17:38087831- 38089396	1566	AAAggatccGAGCCATGAGGTGATAATTATGGAA	AAAgtcgacATGAAAAAGATCACCCTAAATCCCT
rs12709365/rs13380815	chr18:69576417- 69579429	3013	ATTTggatccTGGAAGTTCAGTGAGTGTGTC	TATTgtcgacTCCTCATGCTTCCGGTTGTC

* contained the minor allele (T) of rs12722489 making the sequence non-active as enhancer.

**Table 3 genes-10-00077-t003:** Primers used for PCR mutagenesis.

SNP	Forward primer (5′-3′)	Reverse primer (5′-3′)
rs12946510	GAGTTAAAAATAAAACCACAGCAA	CTGTGGTTTTATTTTTAACTCTGT
rs2313430	GAGATCTTTTTTTCATGTTCTTTTC	GAACATGAAAAAAAGATCTCACTCA
rs4795397	GAAAAGGCCAGTCGGGCTCCATC	TGGAGCCCGACTGGCCTTTTCTG
rs12709365	CCTTGGAACATAGGTATTATTAATTA	ATTAATAATACCTATGTTCCAAGGCA
rs13380815	ATGACAGAATTGAGATTTTGTGGGA	CCACAAAATCTCAATTCTGTCATAT
rs8067378	CGTTATAAATGGGGAAAAACGTT	TTTTTCCCCATTTATAACGTTACA

**Table 4 genes-10-00077-t004:** Electroporation parameters used for transfection.

Cell Line	Pulse Voltage, V	Pulse Width, ms	Pulse Number
MP1	1300	30	1
Jurkat	1350	10	3
Nalm6	1300	30	1
MT-2	1400	30	1
U-937	1400	30	1

**Table 5 genes-10-00077-t005:** Association data for the candidate functional single-nucleotide polymorphisms (SNPs) in the 17q12-21 locus.

Fine-Mapping Study	Candidate SNP	Risk Allele	RAF in European Population (KGph3)	Ancestral/Alternative Allele	Index SNP	GWAS (Ref. #)	Associated Disease	Association *p*-Value	OR
Farh et al., 2015; Schmiedel et al., 2016	rs12946510 (intergenic)	T	0,47	C/T	rs12946510	[39]	Multiple sclerosis	2.90 × 10^−9^	1.07
Schmiedel et al., 2016	rs2313430 (intronic)	T	0,52	T/C	rs2305480	[38]	Ulcerative colitis	3.01 × 10^−8^	1.15
Schmiedel et al., 2016	rs4795397 (intergenic)	G	0,48	A/G	rs2305480	[38]	Ulcerative colitis	3.01 × 10^−8^	1.15
Farh et al., 2015	rs12709365 (intronic)	G	0,47	A/G	rs2872507	[9]	Ulcerative colitis	5 × 10^−11^	1.15
						[10]	Crohn’s disease	5 × 10^−9^	1.12
						[40]	Crohn’s disease	2 × 10^−9^	1.14
						[41]	Type 1 diabetes autoantibodies	2 × 10^−6^	1.10
						[7]	Rheumatoid arthritis	9 × 10^−7^	1.10
					rs12936409	[42]	Rheumatoid arthritis	2.8 × 10^−9^	1.10
Farh et al., 2015	rs13380815 (intronic)	G	0,47	A/G	See above				
Farh et al., 2015	rs8067378 (intergenic)	G	0,51	G/A	rs8067378	[38]	Ulcerative colitis	9.74 × 10^−8^	1.12
					rs8067378	[43]	Primary biliary cirrhosis	6.05 × 10^−14^	1.26

KGph3 = phase 3 of the 1000 Genomes Project [44]; RAF = risk allele frequency; OR = odds ratio; GWAS = genome-wide association study (GWAS).

**Table 6 genes-10-00077-t006:** Association of studied variants with expression levels of genes located within 100 kb distance from the candidate functional SNPs.

Gene	Candidate SNP	Cell Type	Evidence Type *	Reference
STARD3	rs13380815/ rs12709365	primary monocytes (24 h LPS-stimulated; IFNγ-stimulated; naïve)	by LD (r^2^ = 1)	[20]
PGAP3	rs2313430/ rs8067378	primary monocytes (2 h LPS-stimulated)	by LD (r^2^ = 1)	[20]
	rs12709365/ rs13380815	primary monocytes (24 h LPS-stimulated)	by LD (r^2^ = 1)	[20]
	rs2313430/ rs8067378	primary monocytes (IFNγ-stimulated)	by LD (r^2^ = 1)	[20]
	rs12709365/ rs13380815	naïve primary monocytes	by LD (r^2^ = 1)	[20]
IKZF3	rs2313430	naïve primary monocytes	by LD (r^2^ = 1)	[21]
	rs2313430/ rs8067378	whole blood (meta-analysis)	by LD (r^2^ = 1)	[22]
	rs13380815/ rs12709365	whole blood	by LD (r^2^ = 0.967)	[23]
	rs8067378/ rs2313430	whole blood	by LD (r^2^ = 0.875)	[23]
	rs12946510	whole blood	direct data	[24]
ZPBP2	rs2313430/ rs8067378	lymphoblastoid cell lines	by LD (r^2^ = 0.967)	[25]
GSDMB	all	whole blood, spleen, EBV-immortalized B cells (except rs12946510)	direct data	GTEx V7 [50]
	rs8067378	whole blood	direct data	[26]
	rs12709365/ rs13380815	whole blood	by LD (r^2^ = 1)	[26]
	rs8067378	EBV-immortalized B cells	direct data	[27]
	rs2313430/ rs8067378	primary peripheral blood CD4+ lymphocytes	by LD (r^2^ = 1)	[28]
	rs8067378/ rs2313430	whole blood, meta-analysis	by LD (r^2^ = 1)	[22]
	rs8067378	lymphocytes (inferred)	direct data	[29]
	rs2313430/ rs8067378	whole blood	by LD (r^2^ = 1)	[23]
ORMDL3	all	whole blood, spleen, EBV-immortalized B cells	direct data	GTEx V7 [50]
	rs13380815/ rs12709365	whole blood	by LD (r^2^ = 1)	[30]
	rs8067378	EBV-transformed B cell lines	direct data	[30]
	rs13380815/ rs12709365	primary peripheral blood CD4+ lymphocytes	by LD (r^2^ = 0.874)	[28]
	rs8067378/ rs2313430	primary monocytes (24 h LPS-stimulated)	by LD (r^2^ = 0.091)	[20]
	rs8067378/ rs2313430	naïve primary monocytes	by LD (r^2^ = 1)	[20]
	rs4795397	whole blood, meta-analysis	direct data	[22]
	rs8067378	lymphocytes (inferred)	direct data	[29]
	rs2313430/ rs8067378	whole blood	by LD (r^2^ = 1)	[23]
	rs2313430/ rs8067378	lymphoblastoid cell lines	by LD (r^2^ = 1)	[25]
GSDMA	rs8067378/ rs2313430	primary monocytes (2 h and 24 h LPS-stimulated; IFNγ-stimulated)	by LD (r^2^ = 1)	[20]
PSMD3	rs12709365/ rs13380815	primary monocytes (24 h LPS-stimulated)	by LD (r^2^ = 0.874)	[20]

* “Direct data” type of evidence indicates that a candidate SNP was present on the genotyping chip used in the corresponding study. Otherwise, we looked for expression quantitative trait loci (eQTLs) in strong linkage disequilibrium (LD; r^2^ > 0.8 according to the 1000 Genomes Pilot Project [44]) with any of the candidate SNPs. We provide here the variant in the highest LD with an index eQTL, or a pair of variants if they are equally associated. LPS = lipopolysaccharide; IFNγ = interferon gamma; EBV = Epstein–Barr virus.

**Table 7 genes-10-00077-t007:** Cell lines used for transfection.

Name	Origin	Phenotype	References
Nalm6	Pre-B cells cultured from the blood of a patient with non-T, non-B acute lymphoblastic leukemia	B cell precursors	[56]
MP1	EBV-transformed peripheral B lymphocytes	Mature IgM-producing B cells	[57]
Jurkat	Cells derived from the peripheral blood of a patient with acute lymphoblastic leukemia	T helpers	[58,59,60]
MT-2	Human T cell leukemia virus type 1 (HTLV-1) infected leukocytes from cord blood	Regulatory T cells	[61]
U-937	Monocytic cell line derived from diffuse histiocytic lymphoma	Monocytes	[62]

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
