# Peer review of "Functional SNPs in the Human Autoimmunity-Associated Locus 17q12-21"

_genes, 2019, doi:10.3390/genes10020077_

Round 1

Reviewer 1 Report

Ustiugova et al report here four SNPs in the human 17q12-21 locus associated with several important autoimmune diseases (multiple sclerosis, ulcerative colitis, Crohn’s disease, Type 1 diabetes, primary biliary cirrhosis, rheumatoid arthritis). The authors used an intelligent systematic study approach looking for statistical, epigenetic and functional evidence of causality in various cell lines typed. They conclude that rs12946510, rs4795397, rs12709365 and rs8067378 influenced the reporter expression level in leukocytic cell lines. The strongest effect visible in 3 distinct cell types (Nalm6, MP1, activated U-937) was observed for rs12946510, a SNP that modifies gene transcription by altering MEF2A/C and FOXO1 binding sites, both transcription factors needed for B cell development and normal functioning.

The manuscript is interesting, well written in good English and the information shown here is up-to-date. In my opinion the paper might be accepted in its present form.

Author Response

Thank you very much for reviewing and positive comment on our work.

Reviewer 2 Report

The authors in this study have tried to identify the most probable causative autoimmunity‐associated SNPs in the human 17q12‐21 locus by analyzing six SNP candidates via luciferase reporter system. The overall manuscript is well written but needs improvement in their conclusions. My suggestions are the following,

The introduction section could have a paragraph on how auto-immune diseases are triggered. What are the different cell types mainly involved? Please refer to the review article by Raychaudhuri et al 2016 on autoimmune disease and risk alleles. 

Figure 1 could also have an image of the luciferase reporter vector constructed, depicting the promoter and enhancer regions. The resolution of 1 a is low can be improved. Also it is not clear if rs2313430 SNP is within the gene IKZF3 or upstream of the gene. If it is within the gene, the protein could be non functional.

"only rs4795397 showed weak but statistically significant influence on luciferase activity. Its risk allele corresponded to lower luciferase signal in MP1 cells" . The data is not convincing for this statement. What is the p value for the significance?

In the discussion section needs further elaboration on the analysis of rs8067378 SNP. Figure 1 a does not indicate presence of DNAse hypersensitivity or histone modification signatures around this SNP. Then what evidence is there for it to be a putative enhancer region needs to be explained. Reference 17 has shown in their study that because of the SNP, ORMDL3 showed higher expression in immune tissues and has been implicated in many diseases involving dysregulated immune responses. This could be a reason why MT-2 is showing higher expression with the risk allele. 

Lastly further experiments are needed in the future for having a better understanding of the causative variants such as analysis of target protein expression and the immunophenotype of cell lines carrying the SNPs needs to be measured (Can be mentioned in the discussion). Ultimately how the autoimmune disease could manifest because of the potential SNP needs to be explained.

Author Response

Point 1: The introduction section could have a paragraph on how auto-immune diseases are triggered. What are the different cell types mainly involved? Please refer to the review article by Raychaudhuri et al 2016 on autoimmune disease and risk alleles.

Response 1: Thank you for the valuable suggestion. We added the corresponding paragraph to the Introduction.

Point 2: Figure 1 could also have an image of the luciferase reporter vector constructed, depicting the promoter and enhancer regions. The resolution of 1 a is low can be improved. Also it is not clear if rs2313430 SNP is within the gene IKZF3 or upstream of the gene. If it is within the gene, the protein could be non functional.

Response 2: We have added an example plasmid map to the Fig. 1. The resolution of Fig.1a indeed became poor after insertion into the Microsoft Word. However, we uploaded a PDF file containing all the figures in satisfactory resolution upon submission. We have discussed this technical issue with the editorial office and will provide the original figure file along with the revised manuscript. We have also improved Fig.1a appearance in the revised version of the Microsoft Word file.

The variant rs2313430 is located in the last intron of IKZF3. We used the original Genome Browser screenshot with the gene map because it is familiar to most readers. The direction of transcription is indicated by arrows next to gene names. To make SNPs positions more clear to readers, we have explained our gene scheme in the caption and made intronic parts bolder.

We also noticed that the positions of rs13380815 and rs12709365 can be unclear to the readers as well, so we added the corresponding notes to Table 1.

Point 3: "only rs4795397 showed weak but statistically significant influence on luciferase activity. Its risk allele corresponded to lower luciferase signal in MP1 cells" . The data is not convincing for this statement. What is the p value for the significance?

Response 3: Thank you very much for careful reading of our article. The p-value for the difference in luciferase signal between the two alleles of rs4795397 is 0.035. The asterisk sign in Fig. 2 reflecting this was lost at some point during figure preparation. We fixed that.

Point 4: In the discussion section needs further elaboration on the analysis of rs8067378 SNP. Figure 1 a does not indicate presence of DNAse hypersensitivity or histone modification signatures around this SNP. Then what evidence is there for it to be a putative enhancer region needs to be explained. Reference 17 has shown in their study that because of the SNP, ORMDL3 showed higher expression in immune tissues and has been implicated in many diseases involving dysregulated immune responses. This could be a reason why MT-2 is showing higher expression with the risk allele.

Response 4: Figure 1a shows rather large region of the genome and only the strongest peaks are clearly visible on this scale. However, even with this resolution a DHS cluster can be seen at rs8067378, as well as a nice H3K27Ac peak in neutrophils and small peaks in both CD4+ , CD8+, B cells and monocytes. If we zoom in this area the presence of enhancer marks can be readily seen (the zoomed rs8067378 area view can be found in the attached PDF version).

Our previous experience shows that areas with even lower peaks can still demonstrate enhancer properties in a reporter gene assay. We also noticed that peak height is not predictive of enhancer strength in the luciferase system. Farh et al. also assigned to this area a status of active enhancer in several immune cell types.

We decided to include only the overall view of the chromosomic region in order to make the figure and the article more compact, keeping in mind that a reader can take a closer look at the cloned areas in the Genome Browser using the coordinates provided in the Materials and Methods section.

McGovern et al. did not conduct experiments that would show direct influence of rs8067378 genotype on ORMDL3 expression level. They pinpointed ORMDL3 and GSDMB solely based on their proximity to the SNP. They studied the role of ORMDL3 in the Unfolded Protein Response (UPR) using epithelial cells, which is irrelevant to our study. Several other genes can be regulated by rs8067378, and conversely, all our candidates are associated with ORMDL3 expression (Table 2). So we do not feel there is enough ground for discussing this particular gene in the context of rs8067378 functionality in MT-2 cells, especially given that Schmiedel et al. found no interaction of ORMDL3 promoter and the intergenic area harbouring rs8067378 in primary CD4+ T cells (Fig. 5 in Schmiedel et al., 2016).

Point 5: Lastly further experiments are needed in the future for having a better understanding of the causative variants such as analysis of target protein expression and the immunophenotype of cell lines carrying the SNPs needs to be measured (Can be mentioned in the discussion). Ultimately how the autoimmune disease could manifest because of the potential SNP needs to be explained.

Response 5: This work was indeed only a first step of our study. We shall now focus on the most reliable and strong candidate, rs12946510. Our plans include testing differential binding of MEF2C and FOXO1, and, more importantly, creating sub-lines of Nalm6 and MP1 bearing alternative alleles to study how this SNP influences gene expression and cell behavior. We have added a paragraph to the Discussion addressing the ultimate goal of post-GWA studies and methods that might be exploited.

Round 2

Reviewer 2 Report

The authors have efficiently revised the manuscript and it is acceptable for publication.